# WIP1 Inhibition by GSK2830371 Potentiates HDM201 through Enhanced p53 Phosphorylation and Activation in Liver Adenocarcinoma Cells

**DOI:** 10.3390/cancers13153876

**Published:** 2021-07-31

**Authors:** Chiao-En Wu, Ahmed Khairallah Mahdi, Chen-Yang Huang, Chiao-Ping Chen, Yi-Ru Pan, John Wen-Cheng Chang, Jen-Shi Chen, Chun-Nan Yeh, John Lunec

**Affiliations:** 1Division of Haematology-Oncology, Department of Internal Medicine, Chang Gung Memorial Hospital, Linkou Branch, Chang Gung University College of Medicine, Taoyuan 333, Taiwan; 8805017@cgmh.org.tw (C.-E.W.); b9202070@cgmh.org.tw (C.-Y.H.); D000017242@cgu.edu.tw (C.-P.C.); wen1902@cgmh.org.tw (J.W.-C.C.); js1101@adm.cgmh.org.tw (J.-S.C.); 2Liver Research Center, Chang Gung Memorial Hospital, Linkou, Taoyuan 333, Taiwan; panyiru0331@cgmh.org.tw; 3Department of Pathology and Forensic Medicine, College of Medicine, Al-Nahrain University, Baghdad 10006, Iraq; dr.ahmedkhairallah@nahrainuniv.edu.iq; 4Newcastle University Cancer Centre, Bioscience Institute, Medical Faculty, Newcastle University, Newcastle upon Tyne NE2 4HH, UK; 5Department of General Surgery, Chang Gung Memorial Hospital, Linkou Branch, Chang Gung University, Taoyuan 333, Taiwan

**Keywords:** HDM201, GSK2830371, MDM2, WIP1, p53, cholangiocarcinoma

## Abstract

**Simple Summary:**

Patients with advanced intrahepatic cholangiocarcinoma (iCCA) have a very poor prognosis, and no targeted therapy is approved for advanced iCCA. A therapeutic strategy for wild-type p53 cancers is the reactivation of p53 by inhibition of its the negative regulators, MDM2, and WIP1. In the present study, we used HDM201 (an MDM2-p53 binding antagonist) to increase p53 stabilization and upregulate the expression of downstream targets (p21 and MDM2) in RBE and SK-Hep-1 liver adenocarcinoma cell lines. The survival rate and clonogenicity decreased after HDM201 treatment in a dose-dependent manner. Combined treatment with HDM201 and GSK2830371 (WIP1 inhibitor) increased p53 phosphorylation, leading to sustained p53 activation. This combination treatment resulted in G2/M phase arrest and promoted cytotoxicity compared with MDM2 inhibitor monotherapy. Furthermore, increased expression of p53 signaling pathway target genes were identified following combination treatment with HDM201 and GSK2830371, suggesting potential roles for this combination strategy in iCCA therapy.

**Abstract:**

Background: Intrahepatic cholangiocarcinoma (iCCA) is an adenocarcinoma arising from the intrahepatic bile duct. It is the second most common primary liver cancer and has a poor prognosis. Activation of p53 by targeting its negative regulators, *MDM2* and *WIP1*, is a potential therapy for wild-type p53 cancers, but few reports for iCCA or liver adenocarcinoma exist. Methods: Both RBE and SK-Hep-1 liver adenocarcinoma cell lines were treated with the HDM201 (Siremadlin) MDM2-p53 binding antagonist alone or in combination with the GSK2830371 WIP1 phosphatase inhibitor. Cell proliferation, clonogenicity, protein and mRNA expression, cell cycle distribution, and RNA sequencing were performed to investigate the effect and mechanism of this combination. Results: GSK2830371 alone demonstrated minimal activity on proliferation and colony formation, but potentiated growth inhibition (two-fold decrease in GI_50_) and cytotoxicity (four-fold decrease in IC_50_) by HDM201 on RBE and SK-Hep-1 cells. HDM201 increased p53 protein expression, leading to transactivation of downstream targets (p21 and MDM2). Combination with GSK2830371 increased p53 phosphorylation, resulting in an increase in both p53 accumulation and p53-dependent trans-activation. G2/M arrest was observed by flow cytometry after this treatment combination. RNA sequencing identified 21 significantly up-regulated genes and five downregulated genes following p53 reactivation by HDM201 in combination with GSK2830371 at 6 h and 24 h time points compared with untreated controls. These genes were predominantly known transcriptional targets regulated by the p53 signaling pathway, indicating enhanced p53 activation as the predominant effect of this combination. Conclusion: The current study demonstrated that GSK2830371 enhanced the p53-dependent antiproliferative and cytotoxic effect of HDM201 on RBE and SK-Hep-1 cells, providing a novel strategy for potentiating the efficacy of targeting the p53 pathway in iCCA.

## 1. Introduction

Cholangiocarcinoma (CCA) consisting of intrahepatic cholangiocarcinoma (iCCA) and extrahepatic cholangiocarcinoma (eCCA) is a relatively rare but aggressive biliary tract cancer. The prevalence of iCCA is much higher in Asian than in Western countries [1], as hepatitis viruses B and C and liver flukes are the main etiological causes of iCCA in Asia [2,3,4]. iCCA is the second most common primary liver cancer. It has aggressive biological behavior and is typically diagnosed at an advanced stage with poor prognosis [5].

Previous clinical trials have evaluated molecular-targeted therapies in combination with chemotherapy, but most of them showed disappointing results in large-scale phase III studies [6,7,8]. Therefore, gemcitabine-based chemotherapy is still the standard treatment for iCCA [9]. Pemigatinib (an FGFR2 inhibitor) demonstrated activity in previously treated advanced cholangiocarcinoma based on a phase 2 trial and was approved by the US FDA in 2020 [10], but only 10–15% of patients with iCCA have the FGFR2 fusion gene [11]. Therefore, this treatment gap identifies iCCA as a priority for exploring novel therapeutic drugs or regimens for advanced or refractory iCCA [12,13,14].

The tumor suppressor p53 is a transcription factor that regulates a number of genes with a broad range of functions, including DNA repair, metabolism, cell cycle arrest, apoptosis, and cell senescence [15,16]. Impaired wild-type (WT) p53 tumor suppressor function resulting from *TP53* mutation or suppression of p53 by its negative regulators frequently occurs in cancers and may subsequently cause cancer cell development, survival, and proliferation [17]. Therefore, restoration of WT p53 function by inhibiting its negative regulators is a therapeutic strategy currently being explored for cancer treatment. MDM2-p53 binding antagonists (MDM2 inhibitors, MDM2i) block the p53-binding pocket of MDM2 and stabilize p53 by preventing its MDM2-mediated ubiquitylation and degradation. This results in activation of the p53 pathway in p53^WT^ (p53-wild type) rather than p53^MUT^ (p53-mutated) cancer cells, thereby causing cell cycle arrest, apoptosis, and growth inhibition of human cancer cells [18]. HDM201 [19], a new-generation, highly potent and selective MDM2 inhibitor, is under investigation in early clinical trials. Wild-type p53-induced phosphatase 1 (WIP1) encoded by the *PPM1D* gene is a phosphatase that negatively regulates p53 by dephosphorylation (Ser15), resulting in suppression of p53 transactivation activity. GSK2830371, a WIP1 inhibitor (WIP1i) is an allosteric inhibitor of WIP1 phosphatase activity and facilitates the proteasomal degradation of WIP1 [20]. We have previously reported the anti-cancer activity of MDM2i in combination with GSK2830371 in cutaneous melanoma and showed that GSK2830371 potentiates the effect of MDM2i on melanoma cells by increasing the function and stability of p53 in a p53-dependent manner [21]. As more than half of iCCA are WT p53 [13], we investigated the in vitro activity of HDM201 and GSK2830371 in p53^WT^ iCCA to explore the potential of this combination for future clinical trials.

## 2. Results

### 2.1. GSK2830371 Alone Exhibited no Growth Inhibitory Activity

To determine the growth inhibitory activity of GSK2830371 in liver adenocarcinoma cell lines, RBE and SK-Hep-1 cells were treated with GSK2830371 alone at different concentrations up to a maximal concentration of 10 µM, and cell proliferation was measured by CCK-8 assay. No significant growth inhibitory activity was observed for GSK2830371 alone over the range of GSK2830371 concentrations used (Figure 1).

### 2.2. GSK2830371 Potentiated the Growth Inhibitory Activity of HDM201

Synergism between two compounds evaluated by a combination index approach only, has relevance when both agents have a dose-dependent effect [22]. Since GSK2830371 alone showed no dose-dependent effect on cells, potentiation rather than synergy was examined in the following experiments. A fixed concentration of 2.5 µM GSK2830371 with minimal effects alone was used in subsequent combination experiments to evaluate the potentiation of GSK2830371 in combination with HDM201 as in previous reports for other cancers and MDM2i [21,23]. RBE and SK-Hep-1 cells were treated with HDM201 alone or in combination with 2.5 µM GSK2830371 for 72 h and 96 h, and cell proliferation was measured by CCK-8 assay (Figure 2A–D). The % growth inhibition of the combination treatment was normalized to that of GSK2830371 alone. GSK2830371 significantly increased the growth inhibitory and cytotoxic activity of HDM201, as evidenced by deeper growth inhibition and lower GI_50_ for combination treatment than with HDM201 alone (Figure 2E). No growth inhibitory activity was observed for HDM201 alone or in combination with GSK2830371 were observed in the normal cholangiocyte cell line MMNK-1 (Appendix A).

### 2.3. GSK2830371 Enhanced the Inhibition of Colony Formation by HDM201

A clonogenic assay was performed to evaluate the potentiating effect of GSK2830371 on the cytotoxic activity of HDM201 alone in colony formation assays. GSK2830371 alone had minimal effects (<20%) on colony formation in SK-Hep-1 and RBE cells when seeded at low densities. Fewer colonies were formed after combination treatment than with HDM201 treatment alone, showing that the addition of GSK2830371 significantly reduced colony formation when RBE and SK-Hep-1 cells were treated with HDM201 (Figure 3A–D). Similar to the growth inhibition assay, no significant inhibition of colony formation was observed in MMNK-1 cells (Appendix A). As HDM201 alone had inhibition effects in a dose-dependent manner on colony formation, the half-maximal inhibitory concentration (IC_50_) was calculated after normalization against either GSK2830371 or DMSO treatment to minimalize the cytotoxic effects of GSK2830371 alone. A significantly reduced IC_50_ of HDM201 after normalization to either GSK2830371 or DMSO was observed (Figure 3E,F). This indicates that GSK2830371 potentiates both growth inhibition and cytotoxicity induction by HDM201.

### 2.4. GSK2830371 Induced p53 Phosphorylation and Stabilization via WIP1 Inhibition

To confirm the on-target effects of HDM201 and GSK2830371 and explore possible mechanisms, RBE and SK-Hep-1 cells were treated with HDM201, GSK2830371, or a combination of both at 2, 4, 6, and 24 h. Immunoblotting showed that HDM201 stabilized p53, as evidenced by increased p53 detection after 2 h HDM201 treatment. The p53 stabilization persisted for 24 h. MDM2, p21, and WIP1 are direct transcriptional targets of p53, and expression of these proteins increased after 2 h treatment, achieved a peak level at 4 h, and maintained persistently increased expression for 24 h. When the cells were treated with HDM201 in combination with GSK2830371, decreased WIP1 phosphatase protein and correspondingly increased phosphorylated p53 (Ser15) were noted. Although no significant change in total p53 protein was observed after combination treatment, a slight increase in p21 was evident, and of MDM2 at 24 h (Figure 4A). Similar findings were noted when RBE and SK-Hep-1 cells were treated with two doses of HDM201 (0.1 µM and 1 µM) and GSK2830371 (2.5 µM) for 6 h and 24 h. Dose and time-dependent responses were observed using immunoblotting to detect protein and qRT-PCR for mRNA (Figure 4B–E). Combination treatment induced more mRNA and protein expression for both p21 (encoded by CDKN1A) and MDM2 than HDM201 mono-treatment. Significant increases in p21 protein and mRNA were found with combination treatments, particularly after high concentration (1 µM vs. 0.1 µM) and long incubation (24 h vs. 6 h).

### 2.5. GSK2830371 Induced Increased G2/M Arrest When Added to HDM201

RBE and SK-Hep-1 cells were treated with two doses of HDM201 (0.1 µM and 1 µM) and GSK2830371 (2.5 µM) for 24 h and 48 h, and FACS was performed to evaluate the cell cycle distribution after HDM201 and GSK2830371 treatment. Compared to untreated DMSO controls, GSK2830371 alone did not influence cell cycle distribution in either RBE or SK-Hep-1 cells. For RBE cells, a decreased number of cells in S phase and modest G1 accumulation was observed after 0.1 µM HDM201 treatment and following combination treatment with GSK2830371 for 24 h. Increased G2/M phase arrest was induced by treatment with 1 µM HDM201 combined with GSK2830371 for 48 h compared with HDM201 monotherapy (Figure 5A–C). For SK-Hep-1 cells, 0.1 µM HDM201 treatment induced G1 arrest, and 1 µM HDM201 induced G2/M arrest. GSK2830371 further increased the G2/M arrest when combined with HDM201 (Figure 5D–F).

### 2.6. RNA Sequencing Identified Transcript Changes Resulting from Simultaneous Inhibition of MDM2 and WIP1

RNA sequencing was performed to further investigate the mechanism underlying the enhanced effect of combined targeting of MDM2 and WIP1 in RBE cells. RBE cells were treated with 0.1 µM HDM201 with or without 2.5 µM GSK2830371 for 6 h and 24 h, and mRNA was extracted for RNA sequencing. The samples from three independent experiments performed in duplicate were collected for RNA sequencing, and the resulting data were analyzed.

The differentially expressed genes were selected for more than 2-fold changes and *p*-values for differences in expression of less than 0.05. From four different comparisons, HDM201 alone and in combination with GSK2830371 for 6 h and 24 h, 26 consistently altered genes were identified (21 upregulated genes and 5 downregulated genes) (Figure 6A, Appendix A). Besides the genes previously reported to be associated with p53 activation in the literature, downregulation of WDR76 was newly identified and may be a novel target that merits further investigation. RNA sequencing demonstrated potentially novel transcript alterations due to combination treatment with HDM201 and GSK2830371 in addition to transcript level changes of well-known genes with established involvement in p53 signaling pathways, apoptotic processes, and response to DNA damage.

We further selected differentially expressed genes from combined HDM201 and GSK2830371 treatment for 24 h vs. DMSO, detected with a q-value less than 0.01 [24], and performed gene ontology (GO) gene set enrichment analyses. In brief, we selected differentially expressed genes with fold change ≥ 2 and performed GO enrichment using the Database of Immune Cell Expression (DICE) [25]. The results are shown in Figure 6B, using a GO-net with a *p*-value less than 0.001 [26]. Differentially expressed genes from combination treatment were enriched for p53-dependent signal transduction pathway regulated genes involved in the DNA damage response, response to stress, and regulation of apoptotic processes (Appendix A). In addition, differentially expressed genes, analyzed using the KEGG pathway database and gene ontology database for protein–protein interaction (Appendix A) using an adjusted *p*-values less than 0.01, were enriched for genes involved in platinum drug resistance and the p53 signaling pathway (Appendix A) [27,28]. The results of analyses for treatments other than combination for 24 h are shown in Appendix A.

## 3. Discussion

Targeting negative regulators of WT p53 could represent a novel therapeutic strategy for the treatment of advanced BTC. Here, this in vitro study provided initial evidence for the feasibility of such treatment using liver adenocarcinoma cell lines. HDM201, a potent and highly specific second-generation MDM2 inhibitor, stabilized p53 followed by transactivation of its downstream transcriptional targets, leading to cell cycle arrest, growth inhibition, and cytotoxicity. In addition, combined treatment with an inhibitor of the WIP1 phosphatase, GSK2830371, resulted in an increase in the level of phosphorylated p53 protein in cells and enhanced HDM201 activity, producing an increased inhibition of both proliferation and colony formation ability (Figure 7). RNA sequencing demonstrated potentially novel increases in gene expression (discuss in detail below) not previously reported to be involved in p53 signaling pathways, apoptotic processes, and response to DNA damage.

GSK2830371 has been reported to enhance p53-mediated tumor suppression by MDM2 inhibitors, nutlin-3 [23,29], nutlin-3a [30], and RG7388 [23] or by chemotherapy [20,29]. In addition, combination treatment with inhibitors of WIP1 and MDM2, which act via inhibiting p53 dephosphorylation and promoting p53 stabilization, have been reported to result in increased levels of p53 phosphorylation (Ser15), acetylation (Lys382), and increased expression of p53-dependent transcriptional target genes. Furthermore, combination treatment with GSK280371 and an MDM2 inhibitor has been shown to result in increased growth inhibition and cytotoxicity in a wide range of cancer cell types, including melanoma, osteosarcoma, colon carcinoma, neuroblastoma, and breast and ovarian cancer cells [21,23,30]. Here, we explored and validated this combination treatment in liver adenocarcinoma cells. Our results showed that RBE and SK-Hep-1 cells were insensitive to inhibition of proliferation mediated by GSK2830371 alone, but were sensitive to combined treatment with WIP1i and MDM2i through increased stabilization and p53 phosphorylation (Ser15) compared with MDM2i treatment alone. This combination therapy might further trigger an additive or synergistic response through the role of MDM2-mediated and p53-dependent transcription in WIP1 expression, as suggested by Gilmartin et al. [20] and Esfandiari1 et al. [23]. This response was also observed in our results, as indicated by the increased expression of WIP1 after treatment with MDM2i alone (Figure 4A,B,D). Another aspect of the autoregulatory feedback loop involving WIP1, p53, and MDM2 has been described as facilitating the termination of the p53 response after DNA damage [31]. The MDM2-p53 regulatory loop has been reported to be stabilized by the gatekeeper, WIP1, which may dephosphorylate both MDM2 and p53 to promote the MDM2-mediated ubiquitylation and degradation of p53 [32]. Although WIP1i alone can increase p53 stability and activity by inhibiting p53 dephosphorylation, the effect on downstream transcriptional target gene expression appeared modest compared with the antiproliferative and clonogenic inhibitory response to combination treatment (Figure 4B,D). However, it is clear that the potential anti-tumor effect of p53-dependent treatment can be boosted by targeting both WIP1 and MDM2 negative regulators of p53 [20,29,30].

Although p53 regulates cell cycle arrest and apoptosis, the results of p53 reactivation are cancer- or cell line-dependent [21,33]. For example, the primary cellular response to p53 reactivation in murine liver cancers was not apoptosis, but cellular senescence [33]. Apoptosis was also not found in all melanoma cell lines following p53 activation by MDM2 and WIP1 inhibitors [21]. In the current study, apoptotic responses, evaluated by increased FACS Sub-G1 signals and caspase 3/7 activity, were not found in either of the liver BTC cell lines after HDM201 treatment (data not shown), which is similar to the findings in liver cancer cells [33]. Nevertheless, the increased reduction in colony forming ability of the cells following combination treatment indicated that there is a reduced ability of the cells to recover from the inhibition of proliferation.

In the RNA sequencing analysis, 26 genes were identified as potentially either directly or indirectly p53-regulated genes. The transcript alterations included those for genes responsible for cell cycle arrest (*BTG2*, *CDKN1A*), apoptosis (*TP53I3*), ROS control (*FDXR*), DNA repair (*POLH*, *SESN1*), survival (*GPR87*), p53 regulation (*MDM2*), and others (*GDF15*, *CES2*, *ABCA12*, *PLK3*, *CES2*) which have been well established as p53 transcriptional target genes [34,35]. Notably in the context of liver cancer, *CES2* is associated with drug metabolism [36], and *ABCA12* has been associated with hepatic lipid metabolism [37]. Most of the genes found to be upregulated have been frequently associated with regulation by p53 signaling pathways, indicating that the combination of HDM201 and GSK2830371 predominantly involves enhanced activation of p53 function without the involvement of alternative signaling pathways.

Among the upregulated genes identified in this study, some, such as *GRIN2C* and *KLHDC7A,* have been reported in previous studies to be p53 transcriptional targets, but such studies have been limited. [34,38]. *INKA2* has been reported as a novel, direct downstream target of p53 that potentially decreases cell growth by inhibiting the PAK4-β-catenin pathway [39]. *INPP5D* is a hypoxia-induced target gene of p53 [40]. *DGKA*, which is involved in phospholipid signaling and metabolism, was found to be induced by p53 following cisplatin treatment, and has been implicated in radiation induced fibrosis [41,42]. Although *NECTIN4* and *PLXNB3* have been linked with p53, studies are rare [43,44,45,46]. Although *ACTA2* has been reported as a p53 transcriptional target [47], this is an example where the association may be context-dependent [35]. For those above-mentioned genes that have not been commonly reported as p53 transcriptional targets, this is plausibly due to context or cancer type dependence, and they may have particular relevance to BTC.

Regarding the significantly downregulated genes, *CCNE2* [48], *FAM111B* [49], and *ZNF367* [50] have previously been reported to be downregulated by p53. *UHRF1*, an E3 ligase, promotes non-degradative ubiquitination of p53 and suppresses p53-dependent transactivation and apoptosis [51]. *WDR76* is a tumor suppressor that acts via RAS degradation [52]. Neither *UHRF1* nor *WDR76* have been previously reported to be regulated by p53. Future studies will explore the specific genes that have not been previously reported to be regulated by p53 to identify potential novel therapeutic targets or response biomarkers, which may be particularly relevant to liver adenocarcinomas. In recent years, p53-dependent suppression has been attributed to indirect control by the cooperation between p21 and DREAM (dimerization partner, RB-like proteins, E2Fs, and MuvB core), a transcriptional repressor mechanism [50]. Therefore, p53–DREAM would be worth investigating as a mechanism for the downregulated genes.

## 4. Materials and Methods

### 4.1. Cell Lines and Compounds

Since the effect of MDM2 inhibitors is limited to p53^WT^ cancer cells, a panel of BTC cell lines (SSP-25, HuCCT1, SUN-1196, SNU-308, RBE, TFK-1, and TGBC-24TKB) were sequenced for *TP53* and all but RBE [53] were *TP53*-mutated cells (Appendix A). Therefore, we included SK-Hep-1 [54], a liver adenocarcinoma cell line, as a representative p53^WT^ cell line in the current study. These two p53^WT^ liver adenocarcinoma cell lines (RBE and SK-Hep-1) were authenticated and cultured in RPMI medium 1640 (Gibco) and DMEM (Gibco) containing 10% FBS, respectively. MMNK-1, a highly differentiated immortalized human cholangiocyte cell line, was used as a control. HDM201 was obtained from Novartis Pharmaceuticals under material transfer agreement (MTA), and GSK2830371 was purchased from Sigma-Aldrich. All compounds were initially dissolved in DMSO and used to dose cells at a final concentration of 0.5% DMSO, which alone has minimal cytotoxic effects on cells. Original western blots can be found at Appendix A.

### 4.2. Growth Inhibitory Assay

Cells were seeded at 3000/well in 96-well plates overnight before 72 h and 96 h treatment with HDM201, GSK2830371, or a combination of both. CCK8 reagent (Dojindo Molecular Technologies, Inc., Kumamoto, Japan) was used according to the manufacturer’s instructions for measuring cell number and viability, and absorbance optical density (OD) was measured at 450 nm using a microplate spectrophotometer. The concentration of the compound that can inhibit the growth of the cells by 50% compared to the solvent control (GI_50_ value) was determined. The % growth inhibition was calculated as [(T-T_0_)/(C-T_0_)] × 100, where T_0_ is the cell count at day 0, C is the DMSO control OD, and T is the OD at the test concentration [55]. Therefore, if the OD value after treatment is less than the OD at day 0, it produces a negative value, indicating that the compound is not only cytostatic but also cytotoxic.

### 4.3. Clonogenic Assay

Cell lines were seeded in 6-well plates (RBE 800/well and SK-Hep1 200/well) overnight before treatment with HDM201, combined with or without GSK2830371 for 72 h. Fresh medium was replaced, and colonies were allowed to form. The cells were fixed when colonies were visible, depending on their growth rate. The concentration of a compound that can inhibit the number of colonies by 50% compared to DMSO control (IC_50_ value) was also determined.

### 4.4. Immunoblotting

After cells were treated with HDM201 alone and in combination with GSK2830371 for the indicated times, cell lysates were harvested using Pierce™ RIPA Lysis and Extraction buffer (Thermo Scientific, Waltham, MA, USA) with Protease and Phosphatase Inhibitor (Roche, Basel, Switzerland) on ice, and total protein concentration was quantified using a Pierce™ bicinchoninic acid kit (Thermo Scientific). Equal quantities of protein were loaded onto and separated by SDS-polyacrylamide gels. The separated proteins were transferred and immobilized onto Amersham™ nitrocellulose membranes (GE Healthcare Life Science, Chicago, IL, USA). Primary antibodies against p53 (DO-7) (#GTX34938, 1:800, GeneTex, Hsinchu, Taiwan), MDM2 (2A10) (#MABE281,1:1000, Merck Millipore, Burlington, VT, USA), p21WAF1 (12D1) (#2947, 1:1000, Cell Signaling Technology, Danvers, MA, USA), WIP1 (F-10) (#sc-376257, 1:200, Santa Cruz Biotechnology, Dallas, TX, USA), phospho-p53 (Ser-15) (#9284, 1:1000, Cell Signaling Technology), and GAPDH (GT239) (#GTX627408, 1:20000, GeneTex, Hsinchu, Taiwan), and secondary goat anti-mouse/rabbit horseradish peroxidase-conjugated antibodies (#115-035-003/111-035-003, 1:5000, Jackson ImmunoResearch Laboratories, Pennsylvania, PA, USA) were used. All antibodies were diluted in 5% (*w*/*v*) non-fat milk in TBS-T. Proteins were visualized using enhanced chemiluminescence (Merck Millipore, Burlington, VT, USA) and a detector (UVP ChemStudio PLUS, Analytik Jena, Jena, Germany).

### 4.5. RNA Extraction and qRT-PCR

Total RNA was extracted using TRI Reagent (Sigma, St. Louis, MO, USA). RNA purity and concentration were estimated using an ND-1000 spectrophotometer (NanoDrop Technologies, Thermo Scientific, Waltham, MA, USA). Complementary DNA was generated using the HiScript ITM First Strand cDNA Synthesis Kit (Bionovas, Toronto, ON, Canada), as described by the manufacturer. qRT-PCR was carried out using SYBR green RT-PCR master mix (Life Technologies, Carlsbad, CA, USA) according to the manufacturer’s guidelines and the following primers: MDM2: F-AGTAGCAGTGAATCTACAGGGA, R-CTGATCCAACCAATCACCTGAAT; CDKN1A: F-TGTCCGTCAGAACCCATGC, R-AAAGTCGAAGTTCCATCGCTC; GAPDH: F-GTCTCCTCTGACTTCAACAGC, R-ACCACCCTGTTGCTGTAGCCAA. qRT-PCR reactions using 25 ng of cDNA samples per 20 µL final reaction volume, with the standard cycling parameters were performed and products detected in real time on a QuantStudio™ 5 (QS5) system (Applied Biosystems, Thermo Fisher Scientific, Waltham, MA, USA). GAPDH was used as endogenous control and samples of cells exposed to DMSO solvent control were used as a calibrator for each independent repeat. Analysis was carried out using the QuantStudio™ Design and Analysis Software (Thermo Scientific, Waltham, MA, USA).

### 4.6. Fluorescence-Activated Cell Sorting (FACS)

After cells were incubated with HDM201 alone and in combination with GSK2830371 for 24 and 48 h, both floating and adhered cells were pooled and fixed using 70% cold ethanol. Samples were incubated in PI/RNase Staining Solution (Cell Signaling) for 20 min in the dark at room temperature, and then analyzed on a FACSCalibur^TM^ flow cytometer (Becton Dickinson, Oxford, UK) using CellQuest Pro software (Becton Dickinson, Oxford, UK). Doublet discrimination, to distinguish single cells featuring double the normal amount of DNA, was performed by plotting FL2-Area vs FL2-Width. Cell cycle distribution based on DNA content was determined using FlowJo VX software (Becton Dickinson, Oxford, UK).

### 4.7. RNA Sequencing

Total RNA 200 ng from each cell line sample was prepared for sequencing using Illumina TruSeq Stranded mRNA Sample Preparation Kit. Fifteen-cycle-PCR was performed, and the libraries were quantified using the LC480 qPCR system (Roche). Then, all libraries were pooled into a new 1.5 mL tube as a 10 nM pooled DNA library. The final pool was used for sequencing (illumina NextSeq sequencer, 2 × 150 bp). The raw output file of each individual library was > 3GB. The sequence of each read was trimmed based on the quality score (Q30), and read lengths less than 45 bp were discarded prior to further analysis [56]. Finally, reads were aligned to the human hg19 reference genome using STAR aligner [57] for subsequent comparison and analysis. The datasets are available at Gene Expression Omnibus, data series GSE179787.

### 4.8. Statistical Analysis

Data are presented as mean + standard error of the mean (SEM) unless otherwise stated. Statistical tests were carried out using GraphPad Prism 6 software, and all *p*-values represented unpaired t-tests of at least three independent repetitions. A *p*-value of less than 0.05 was considered statistically significant.

## 5. Conclusions

In conclusion, the current study demonstrated that combination treatment with MDM2 and WIP1 inhibitors is a potential novel strategy for liver adenocarcinoma or iCCA. As more than half of the BTCs are WT p53, future clinical trials and exploration are recommended. Novel genes identified in the current study after p53 reactivation should be investigated for further understanding of p53 regulation in iCCA as well.

## Figures and Tables

**Figure 1 cancers-13-03876-f001:**
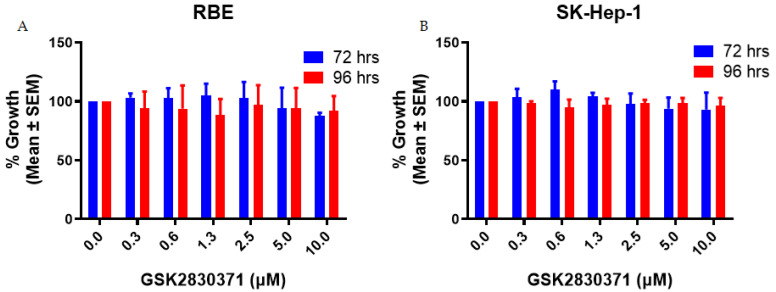
GSK2830371 alone exhibited minimal growth inhibitory activity. RBE (**A**) and SK-Hep-1 (**B**) cells were treated with 0–10 µM GSK2830371 (WIP1 inhibitor) in 0.5% DMSO for 72 and 96 h. Minimal growth inhibitory activity was observed for GSK2830371 treatment alone. All data represent the mean ± SEM from three independent experiments performed in duplicate.

**Figure 2 cancers-13-03876-f002:**
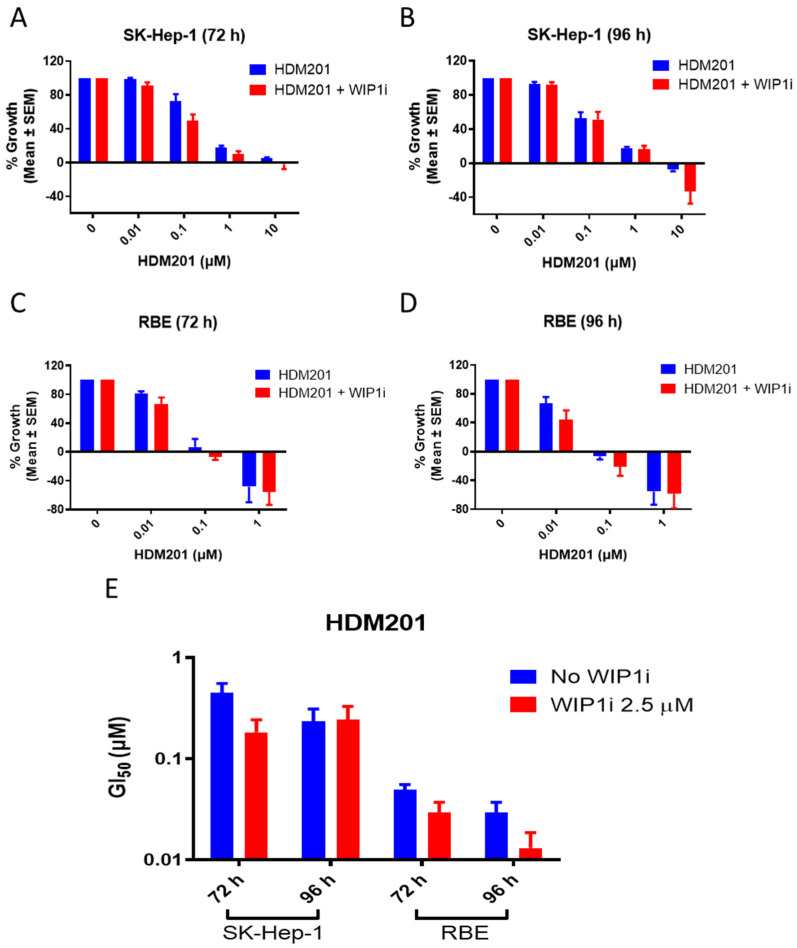
GSK2830371 potentiated the growth inhibitory activity of HDM201. SK-Hep-1 (**A**,**B**) and RBE (**C**,**D**) cells were treated with 0–10 µM HDM201, either alone or in combination with 2.5 µM WIP1 inhibitor (WIP1i) in 0.5% DMSO for 72 (**A**,**C**) or 96 (**B**,**D**) h. The growth rate of HDM201 + WIP1i was normalized against that for cells treated with 2.5 µM WIP1i for 72 and 96 h, respectively. (**E**) Summary of half-maximal growth inhibitory concentration (GI_50_) for HDM201 alone or in combination with WIP1i. GI_50_ values were normalized against those for WIP1i. All data represent the mean ± SEM from three independent experiments performed in duplicate.

**Figure 3 cancers-13-03876-f003:**
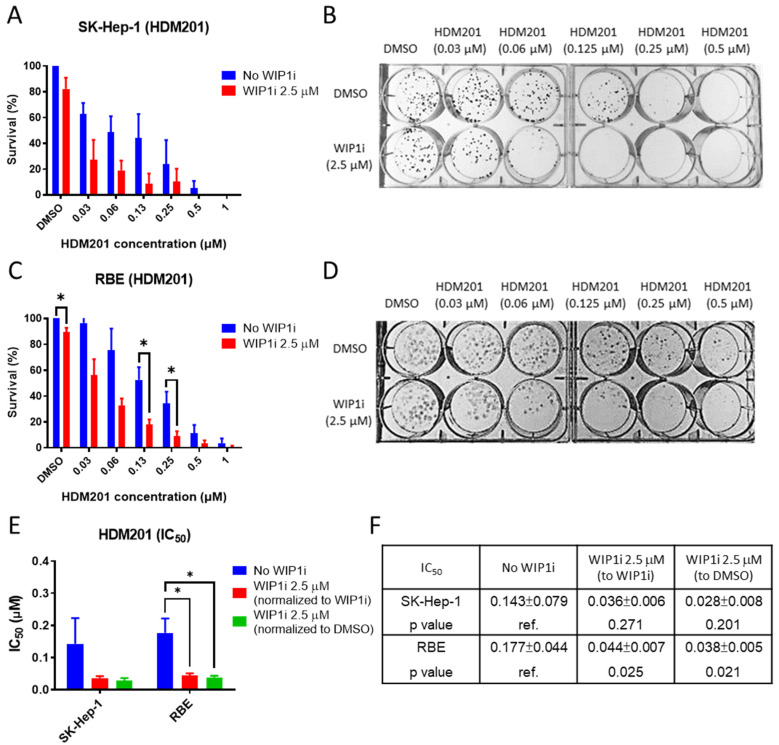
GSK2830371 enhanced colony formation inhibition by HDM201. SK-Hep-1 (**A**,**B**) and RBE (**C**,**D**) were treated with HDM201 alone or in combination with 2.5 μM WIP1 inhibitor (WIP1i) for 72 h and percentage of survival was normalized to DMSO. (**E**,**F**) Summary of IC_50_ (µM) of HDM201 alone or in combination with WIP1i and IC_50_ was normalized to DMSO or WIP1i, respectively. *, *p* < 0.05. Panels **A**, **C**, **E**, and **F** show the mean ± SEM from three independent experiments performed in duplicate.

**Figure 4 cancers-13-03876-f004:**
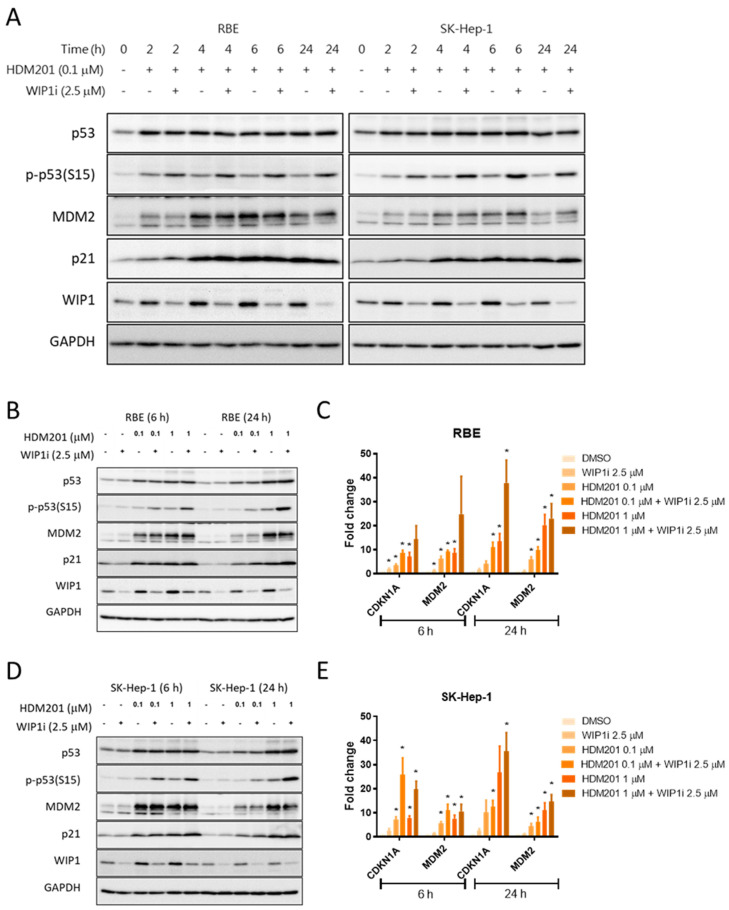
GSK2830371 induced p53 phosphorylation and stabilization via WIP1 inhibition. (**A**) Immunoblotting of RBE and SK-Hep-1 were treated with 0.1 μM HDM201 alone and in combination with 2.5 μM WIP1 inhibitor (WIP1i) for the indicated times. The protein expression of MDM2, WIP1, p53, p-p53, and p21 after treatment with HDM201 alone or in combination with 2.5 µM WIP1i for 6 and 24 h in RBE (**B**) and SK-Hep-1 (**D**) cells. The mRNA expression levels of CDKN1A and MDM2 increased in a dose-dependent manner following treatment with HDM201 or HDM201 + WIP1i for 6 and 24 h in RBE (**C**) and SK-Hep-1 (**E**) cells. * *p* < 0.05. Panels C and E show the mean ± SEM from three independent experiments performed in duplicate.

**Figure 5 cancers-13-03876-f005:**
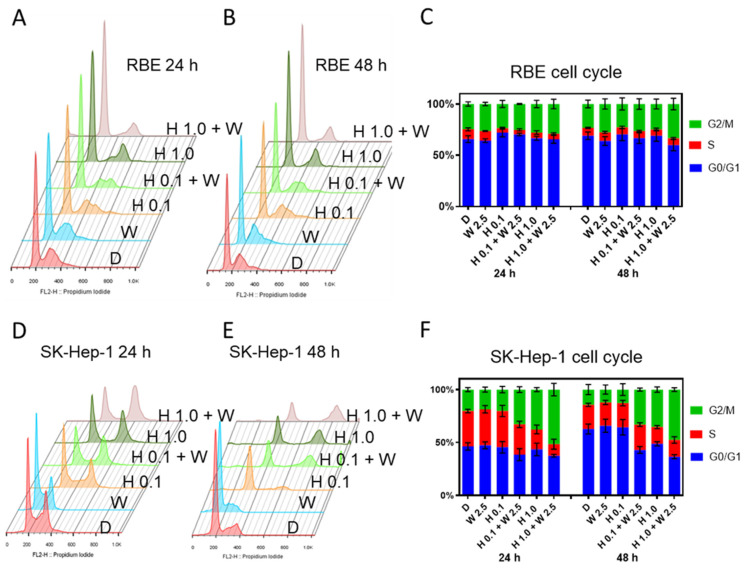
GSK2830371 induced G2/M phase arrest compared with HDM201 alone. RBE (**A**–**C**) and SK-Hep-1 (**D**–**F**) cells were treated with 0.1 or 1 µM HDM201 alone or in combination with 2.5 µM WIP1i for 24 and 48 h. Panels **A**, **B**, **D**, and **E** show representative histogram profiles of the cell cycle. Panels C and F summarize the cell cycle distributions for GO/G1, S, and G2/M phase from at least 3 independent replicates. D, DMSO; W, WIP1i; H, HDM201. All data represent the mean ± SEM from three independent experiments performed in duplicate.

**Figure 6 cancers-13-03876-f006:**
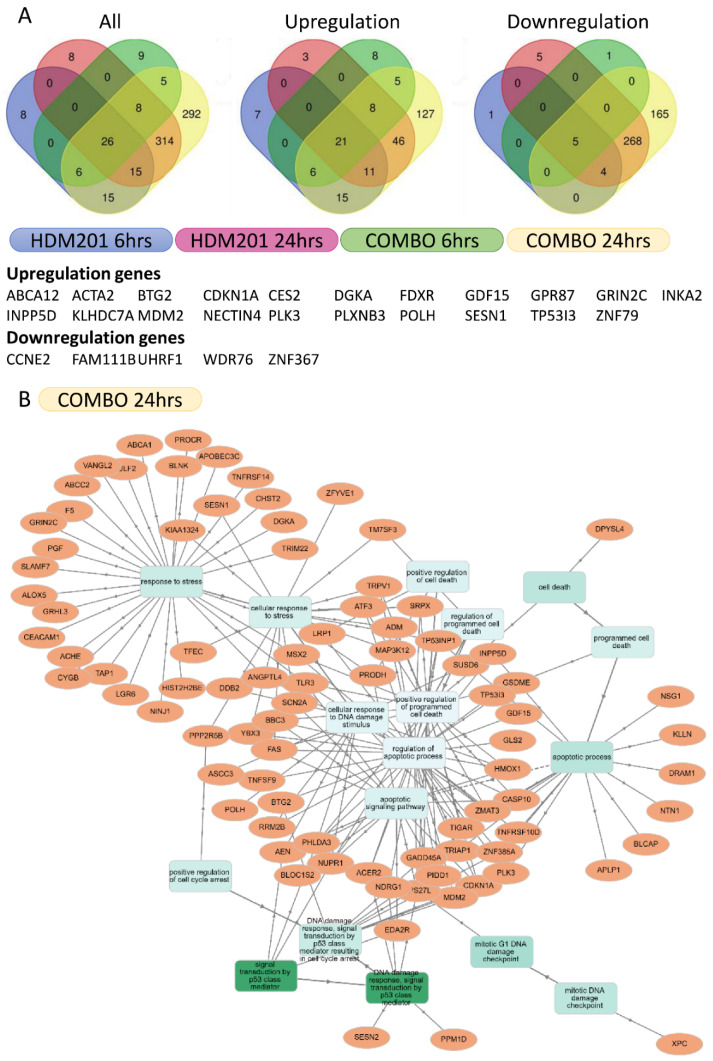
Comparative transcriptomic analyses between treatment with HDM201 alone and in combination with WIP1i (COMBO) for 6 and 24 h in RBE cells. (**A**) Venn diagram of all (left), upregulated (middle), and downregulated (right) differentially expressed genes, as selected according to a >2-fold change and *p* < 0.05. (**B**) REB was treated with HDM201 and GSK2830371 for 24 h. Seventy-seven differentially expressed genes with fold change ≥2 were analyzed by gene ontology (GO) enrichment analysis with Database of Immune Cell Expression (DICE) database using GO-net.

**Figure 7 cancers-13-03876-f007:**
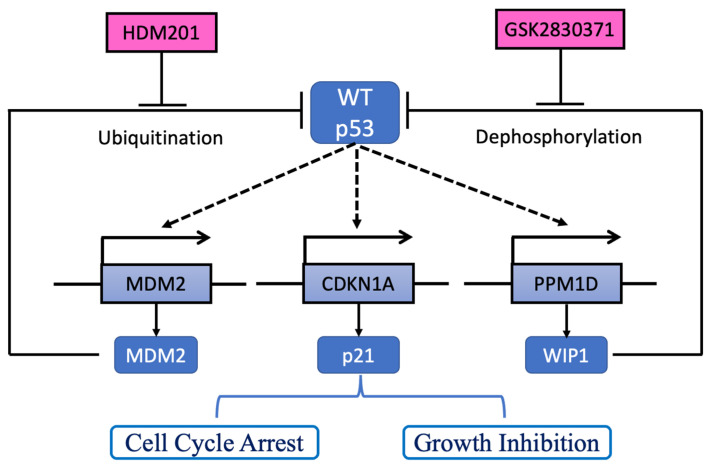
Proposed model for GSK2830371 (WIP1i) potentiation of the growth-inhibitory effects of HDM201. p53 transactivated targets (dashed lines), *MDM2* and *PPM1D* encoding MDM2 and WIP1, which, respectively, suppress p53 function via ubiquitination and dephosphorylation. In addition, p53 transactivates other targets such as *CDKN1A* encoding p21^WAF1^ which induces cell cycle arrest and growth inhibition. MDM2 inhibitor, HDM201, blocks the p53-MDM2 binding interaction, resulting in p53 stabilization. WIP1 inhibition (WIP1i), GSK2830371, increases phospho-p53 (Ser 15) through allosteric inhibition of WIP1 phosphatase. Both inhibitors work on p53 and together achieve greater growth inhibition and cell killing. Dashed lines indicate p53 transcriptional upregulation of the corresponding genes.

## Data Availability

The datasets of RNA sequencing are available at Gene Expression Omnibus, data series GSE179787 (https://www.ncbi.nlm.nih.gov/geo/query/acc.cgi?acc=GSE179787, accessed on 9 July 2021).

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
