# Peer review of "WIP1 Inhibition by GSK2830371 Potentiates HDM201 through Enhanced p53 Phosphorylation and Activation in Liver Adenocarcinoma Cells"

_cancers, 2021, doi:10.3390/cancers13153876_

Round 1

Reviewer 1 Report

The authors have satisfactorily revised the manuscript based on my comments.

Author Response

We thank the Reviewer for your positive and constructive assessment of our manuscript.

Reviewer 2 Report

Dear authors,

thank you for answering my questions and addressing recommendations in your revised version of the manuscript.
However, there are still some point of criticism that have to be addressed before the manuscript is suitable for publication.
1) The statement in line 16 - 17 is still wrong ("A therapeutic strategy for wild-type p53 cancers is reactivation of the negative regulators p53, MDM2, and WIP1").
2) A lot of statements in your manuscript require more linguistic/scientific accuracy, e.g. line 18: "we used HDM201 [...] to increase p53 activation" MDM2 inhibition leads to decreased p53 degradation but activation is not or only indirectly affected. Such statements are also found in lines 209-211 (neither WIP1 nor MDM2 phosphorylate p53), line 220 ( MDM2-mediated proteolysis is totally wrong), line 221 (WIP1i inhibit p53 dephosphoralation but do not mediate p53 phosphorylation) ; line 30: "[...] but little or no reports for iCCA or liver adenocarcinoma exist." either there are few reports or not, but both is impossible.
Please carefully check your manuscript for such false statements.
3) The title of the results section 2.6 ("RNA sequencing identified the mechanism of simultaneously inhibiting MDM2 and WIP1) does not fit. RNA sequencing does not give any mechanistic explanations but only hints for mechanisms.
4) Results section 2.5: You still wrote that there is an G1 arrest in RBE cells after treatment with 0.1 µM HDM201 as well as a G2/M arrest in RBE cells after treatment with 1 µM HDM201 plus GSK2830371. I really do not think that this conclusion is supported by the data you show in figure 5. For RBE cells I do not see an effect of your treatments on the cell cycle distribution. Besides this, the cross reference in line 128 is wrong.
5) The discussion of your RNA sequencing data is still full of mistakes. I did not check all your statements but the few a checked were enough to find false statements:
line 242-243: Diacylglcerol kinase 4 alpha expression has been linked already with p53 (https://doi.org/10.1038/sj.onc.1208755)
line 243 - 244: ACTA2 is a known transcriptional target of p53 (doi:10.1038/sj.onc.1201645)
Please carefully review your candidates and check the literature.

Author Response

Responses to Reviewer 2 Comments:

Dear authors,

Thank you for answering my questions and addressing recommendations in your revised version of the manuscript.

However, there are still some points of criticism that have to be addressed before the manuscript is suitable for publication.

We thank the reviewer for these helpful suggestions regarding our manuscript. All the changes have been highlighted and the point-to point responses to comments are listed below. In addition, the corresponding author, John Lunec, who is a professor in Newcastle University, UK has reviewed whole manuscript and revised some confused points to make it more linguistically and scientifically accurate.

1) The statement in line 16 - 17 is still wrong ("A therapeutic strategy for wild-type p53 cancers is reactivation of the negative regulators p53, MDM2, and WIP1").

Thank you for your comments. We have corrected the statement in revised statement.

2) A lot of statements in your manuscript require more linguistic/scientific accuracy, e.g. line 18: "we used HDM201 [...] to increase p53 activation" MDM2 inhibition leads to decreased p53 degradation but activation is not or only indirectly affected. Such statements are also found in lines 209-211 (neither WIP1 nor MDM2 phosphorylate p53), line 220 (MDM2-mediated proteolysis is totally wrong), line 221 (WIP1i inhibit p53 dephosphoralation but do not mediate p53 phosphorylation); line 30: "[...] but little or no reports for iCCA or liver adenocarcinoma exist." either there are few reports or not, but both is impossible.

Please carefully check your manuscript for such false statements.

Thank you for your comments. We have checked and appropriately modified these and other statements in the revised manuscript (e.g., lines 6-8, 16, 192, and 203-204).

3) The title of the results section 2.6 ("RNA sequencing identified the mechanism of simultaneously inhibiting MDM2 and WIP1) does not fit. RNA sequencing does not give any mechanistic explanations but only hints for mechanisms.

Thank you for this comment. The title of section 2.6 has been changed to be more appropriate and specific.

4) Results section 2.5: You still wrote that there is an G1 arrest in RBE cells after treatment with 0.1 µM HDM201 as well as a G2/M arrest in RBE cells after treatment with 1 µM HDM201 plus GSK2830371. I really do not think that this conclusion is supported by the data you show in figure 5. For RBE cells I do not see an effect of your treatments on the cell cycle distribution. Besides this, the cross reference in line 128 is wrong.

Thank you for your comments. We have modified the statement for RBE cells in section 2.5 to say “For RBE cells, a decreased number of cells in S phase and modest G1 accumulation was observed after 0.1 µM HDM201 treatment and following combination treatment with GSK2830371 for 24 hours’’.

5) The discussion of your RNA sequencing data is still full of mistakes. I did not check all your statements but the few a checked were enough to find false statements:

line 242-243: Diacylglcerol kinase 4 alpha expression has been linked already with p53 (https://doi.org/10.1038/sj.onc.1208755)

line 243 - 244: ACTA2 is a known transcriptional target of p53 (doi:10.1038/sj.onc.1201645)

Please carefully review your candidates and check the literature.

Thank you for your comments. We have rewritten the section describing the RNA sequencing data in the discussion.

This manuscript is a resubmission of an earlier submission. The following is a list of the peer review reports and author responses from that submission.

Round 1

Reviewer 1 Report

The authors have demonstrated that a WIP inhibitor and MDM2 inhibitor's combination treatment has a stronger growth inhibitory potential in liver adenocarcinoma cells. The study demonstrated that the WIP inhibitor (GSK2830371) potentiates an MDM2 inhibitor (HDM201) by enhancing the p53 phosphorylation. The study's observations are based on in-vitro experiments, and it will be interesting if authors could demonstrate the growth inhibitory potential of the combination treatment through in vivo experiments in their future studies. The in vivo models will help in understanding the clinical potential of this combination treatment.

Comments to the authors:

  1. The authors have performed paired t-tests for statistical analysis of the data; however, none of the experiments satisfy the paired t-test analysis requirements. The authors should have performed unpaired t-tests. If it was not a writing error, the authors should reanalyze the data and update the entire data with correct p-values.
  2. The first few sentences in the abstract and summary are repetitive. The authors should consider rewriting the summary of the article.
  3. Figure 5 has not been referenced within the text.
  4. Lines 154-163- The paragraph is difficult to understand, and therefore, authors should consider rewriting it.
  5. RNA sequencing data should be made publicly available. The authors should consider submitting the RNA seq data to the GEO Omnibus database or any other publicly available database and incorporate that information within the manuscript.
  6. The authors should replace "G2 arrest" with "G2/M" arrest.
  7. The authors should provide the dilution of the antibodies used for the immunoblotting.

Reviewer 2 Report

The authors of the manuscript entitled "GSK2830371 Potentiates HDM201 Through Enhanced p53 Phosphorylation and Activation in Liver Adenocarcinoma Cells" investigated the effects of treatment with the WIP1 inhibitor GSK2830371 in combination with the MDM2 inhibitor HDM201 on p53 stability and activation in the p53 wild-type liver adenocarcinoma cell lines RBE and SK-Hep-1. The authors report that GSK2830371 alone did hardly effect the growth of both cell lines but markedly augmented HDM201 mediated growth inhibition due to cell cycle arrest in combination treatment. Moreover, their data implicate that cell cycle arrest is mediated by enhanced p53 activity caused by combination treatment with GSK2830371 and HDM201, which inhibit p53 degradation and dephosphorylation.

TP53 represents a major tumor suppressor, therefore, interfering with its negative regulators in p53 wild-type tumor cells is a very reasonable and promising treatment strategy in various tumor entities.
Since some of the authors listed in the presents manuscript already published at least two studies using a very similar experimental setup but different tumor cell lines, I do not think that the present manuscript is very innovative. Nevertheless, acknowledging the importance of tumor heterogeneity, I think it is reasonable to study the present combination treatment in different tumor entities, in case the presented data are conclusive, well described and discussed.

However, in order to make the present study such a conclusive and well interpreted one, the authors should address following issues:

Major Revisions:
1) The data presented in Figure 3A+C suggest that GSK2830371 at a concentration of 2.5 µM without HDM201 has an effect on clonogenicity of SK-Hep-1 and RBE cells when seeded at low density. Notably, this effect, which is even marked as statistically significant for RBE cells, isn't even commented in the manuscript. Therefore, I think the data/experimental setup presented in Figure 1 is not sufficient to exclude an effect of GSK2830371 as a single agent.
The authors should address this point by providing additional data for GSK2830371 treatment and/or at least discussing the present data, e.g. why does interfering with WIP1 is way less effective than blocking MDM2-p53 interaction? Are there regulators that might compensate for WIP1 blockade?
Besides, the headline of chapter 2.1 does not fit to the data, because obviously there is no affect on the growth of RBE and SK-Hep-1 cells by the applied GSK2830371 concentrations at least in the present experimental setup.

2) I think the data evaluation performed in Figure 2 A, B, C + D is either problematic or poorly described. As far as I see (because of the missing error bars that are clearly present in Figure 1), the authors normalized results for CCK8 assays with increasing HDM201 concentrations to the respective values obtained by treatment with either only DMSO (if cells treated with 0 µM HDM201 were treated with DMSO as a control, which is not stated in the figure legend) or only 2.5 µM WIP1i. Afterwards, they performed a statistical comparison between treatments +/- WIP1i and various HDM201 concentration although these data result from normalization to different controls. From a biological point of view this is not dramatic if the respective controls are almost equal (which is probably the case in this figure), but I would highly recommend to perform normalization to a single control. The proper control for normalization in the present case is "0µM HDM201 without WIP1i". Moreover, the authors should describe any forms of normalization in the figures legends, which is not clearly done in Figure 2.

3) In Figure 3 the authors present data from colony formation assays. However, colony formation assays do not assess "clonogenic killing" or cell death but the potential of single cells to form a colony, which is relevant in the context of metastasis formation. If cell death would be assessed one would have to show that a single cell that has attached to the culture dish after seeding is lost at the end of the experiment due to cell death induction, which is not feasible by the assay procedure. Moreover, normalization process performed in Figure 3E+F again is confusing. As stated in comment (2), the blue and green bars in Figure 3E refer to the proper control.

4) Taking into account the value distribution presented in Figure 5C, I do not see the G1 or G2 arrest described by the authors. In my opinion this statement is only supported by the flow cytometry data for SK-Hep-1 but not RBE cells. Please comment and review the description of the cell cycle data in terms of descriptive and language accuracy.

5) Overall, all figure legends should be revised to ensure that each figure can be interpreted by just reading the respective figure legend. Most of the figure legends in the present manuscript lack important information like the number of performed biological replicates (e.g. line 188: "[...] results of three triplicates" what does this mean? Three biological replicates consisting of each three technical replicates?) and normalization procedure. Moreover, language style has to be revised.

6) Besides some major language-based inaccuracies:
Just some examples
a) line 221/222: "[...] GSK2830371 targeting of WIP1 phosphatase increased p53 phosphorylation and enhanced HDM201 activity"
b) line 228/229: "[...] GSK2830371 potentiates MDM2 inhibitors via p53 phosphorylation"
The discussion comprises several false statements, like in lines 250/252 "GRINC2C and KLHDC7A, both upregulated > 2-fold in this study, may be novel targets, but no further reports are present in the literature [32]."
Both have been reported earlier as potential p53 by:
Allen, M. A., Andrysik, Z., Dengler, V. L., Mellert, H. S., Guarnieri, A., Freeman, J. A., ... & Espinosa, J. M. (2014). Global analysis of p53-regulated transcription identifies its direct targets and unexpected regulatory mechanisms. Elife, 3, e02200.

Overall, the discussion deals with the sparesly described RNA Seq data, which reads in the third paragraph of chapter 2.6 like a material & methods part, rather than the main in vitro results presented in the manuscript. Therefore, I really encourage the authors to rewrite their discussion.

Minor Revisions:
1) Line 16: "A novel treatment for wild-type p53 cancers is reactivation of the negative regulators of p53, [...]" This is definitely not the case.
2) Poor image quality of Figure 3D, Figure 6B and Supplementary Figure S1 has to be improved.
3) Line 136/138: "Significant increases in p21 protein and mRNA were found [....]" Significance not indicated in the respective graphs of Figure 4.
4) Are the histograms depicted in Figure 5A, B, D + E from the same experiment? Because some histograms are markedly shiftet, e.g. histograms of treatments "DMSO" and "WIP1i". Please comment.
5) Ethanol permeabilization and propidium iodide at high concentrations causes increased risk of cell clustering. Have you performed doublet discrimination in flow cytometric analyses? Some G2/M populations show double peaks located at the 2-fold intensity of the G0/G1 peak. Please comment and state in the respective chapter 4.6.
6) Please perform language-editing (especially in chapter "2. Results")

Reviewer 3 Report

In this manuscript, the authors described their studies about the effects of MDM2 and WIP1 inhibitors on RBE and SK-Hep-1 cell lines.  The main finding is that WIP1 inhibitor can improve the efficacy of MDM2 inhibitor.  Although the authors performed a good amount of experiments including RNA sequencing, the results are all in vitro and still preliminary.  The manuscript is not presented well.  The main findings are lack of novelty, and can not be fully supported by limited data showed in the manuscript.  

  1. Since HDM201 and GSK2830371 are already known inhibitors for p53 suppresors, it is not surprising that the authors can see changes in p53 pathway and gene expression in their cellular models.
  2. Although the co-treatment of HDM201 and GSK2830371 caused higher decrease of cell viability, it can be a universal event.  The authors need to test another non-tumor hepatic cell line.
  3. An in vivo model should be tested to see whether co-treatment of HDM201 and GSK2830371 can inhibit tumor growth.
  4. RNA sequencing is not informative in this study.  The authors should further explore the results and identify new factors which may contribute to the observation in cells co-treated with HDM201 and GSK2830371.